# GEAR-FEN: Generalized Feature Representation for Kinematic Human Activity Recognition

## Abstract

This study addresses the challenge of efficient human activity recognition (HAR) with limited training data. We propose GEAR-FEN (**Ge**neralized **A**ctivity **R**ecognition **Fe**ature Extraction **N**etwork), a novel transfer learning (TL) method that transforms kinematic motion signals into a generalized feature space. GEAR-FEN potentially outperforms the state-of-the-art in scenarios with limited training data. This was demonstrated through an evaluation across 11 public HAR datasets (encompassing number of activities ranging from 6 to 33 and number of samples per activity ranging from 8628 to 1140258), using a deep learning model based on convolutional neural networks (CNN), residual bi-directional long short-term memory (ResBiLSTM), and an attention mechanism. Furthermore, we established the generalizability of our method through performance comparisons on an independent dataset encompassing a distinct population and diverse kinematic modalities for 8 activities, and 26121 samples per activity. These findings highlight the potential of our proposed approach in robust feature representation for HAR tasks with limited data sizes.

## 1 Introduction

Human activity recognition (HAR) is fundamental to comprehending and interpreting behavioral patterns in everyday life, allowing for a deeper insight into the intricacies of human behavior Arshad et al. (2022). Recent decades have witnessed a significant evolution in HAR technology, underscored by its growing importance in patient monitoring and rehabilitation, promoting safety and independence for individuals across various populations (Ariza-Colpas et al., 2022; Meng et al., 2020). The advent of wearable sensors, such as inertial measurement units (IMUs), has further fueled this progress, especially for ambulatory and real-life scenarios Qiu et al. (2022).

HAR faces significant challenges due to the inherent variability of motion patterns Chen et al. (2021). This variability arises from: 1) the specific activity set being classified Trabelsi et al. (2022); 2) demographic factors influencing movement patterns, necessitating population-specific models; and 3) variations introduced by the composition of the data collection sensory setup. These factors collectively contribute to the limited generalizability and data-specificity of HAR models. While deep learning architectures have revolutionized HAR by automating feature extraction and superior performance, they have also exacerbated the data dependency challenge due to their inherent complexity (Jain & Kanhangad, 2018; Presotto et al., 2023; Yuan et al., 2024). State-of-the-art (SOTA) models relying on Convolutional Neural Networks (CNNs) and Long Short-Term Memory (LSTM) networks Xia et al. (2020), require vast amounts of labeled data for optimal performance. Bidirectional LSTM (BiLSTM) models have further refined the encoding of temporal information within activities, yet have also increased data dependency Luwe et al. (2022). Recent advancements, such as the incorporation of deep reverse transformer-based attention mechanisms, address the challenge of extracting both global temporal and local spatial features through enhanced feature fusion Pramanik et al. (2023). The ConvTransformer model, which merges CNNs with Transformers, exemplifies this trend by focusing on key features to achieve improved HAR performance Zhang et al. (2023b). Consequently, these advancements have intensified the data dependency issue, presenting a significant bottleneck, as collecting large datasets is resource-intensive and time-consuming, particularly for novel activity sets or populations.

A prevalent approach to addressing data scarcity in HAR is transfer learning (TL), defined as a set of methodologies leveraging knowledge acquired from a related task and domain to enhance performance on the target task Dhekane & Ploetz (2024). In this study, we address the data dependency and scarcity challenge by leveraging a novel generalized TL approach. To summarize, our contributions are as follows:

- We introduce the **g**eneralized **a**ctivity **r**ecognition **fe**ature **e**xtraction **n**etwork (GEAR-FEN). Trained on a collection of diverse HAR datasets, GEAR-FEN is designed to extract representative features from the kinematic time series inherent to human motion.
- We evaluate our proposed method, which combines GEAR-FEN for feature extraction with dataset-specific feature learning networks (FLNs) for classification learning. This approach demonstrably improves classification performance, especially for datasets with limited samples, compared to baseline methods and SOTA.
- We assess our method using an independent, unseen dataset from a novel population with a novel set of activities. This analysis confirms that the features learned by GEAR-FEN effectively represent the underlying human motions and activities.

## 2 RELATED WORK

Within wearable HAR, TL plays a crucial role in adapting predictive models to diverse and dynamic conditions. Strategic approaches, such as heterogeneous transfer, personalized TL, multi-source TL, and task-specific methods, effectively address challenges like sensor position variability, user-specific data adaptation, and the integration of heterogeneous datasets Dhekane & Ploetz (2024). Heterogeneous transfer enhances model reliability across various sensor modalities and positions, while personalized TL focuses on tailoring models to individual characteristics by fine-tuning based on personal activity patterns and physiological data. Multi-source TL exploits the diversity of multiple datasets to build robust models that generalize across novel environments and tasks. Additionally, adapting models to handle varying definitions and labels of activities helps accommodate significant discrepancies across datasets, ensuring the models' applicability in real-world scenarios.

Recent advancements in HAR using wearable data have explored various methodologies to address challenges like data scarcity, sensor diversity, and robust feature representation. One notable recent study implemented multi-task self-supervised learning with a deep convolutional neural network (ResNet-V2, 18 layers) on an extensive unlabeled dataset from the UK Biobank, encompassing 700,000 person-days of accelerometer data, to recognize human activities Yuan et al. (2024). This network was pre-trained and subsequently evaluated via TL across eight public datasets to assess representation quality. However, the study encountered limitations such as the dataset's demographic homogeneity—predominantly Caucasian data from the UK—which could hinder the generalizability of the findings. Additionally, the reliance solely on accelerometer data pointed to the necessity of integrating multimodal sensor data to improve the model's robustness and applicability. Concerns were also raised about the self-supervised learning methods not achieving high-quality representations with free-living activity data, indicating a need for further methodological refinement and exploration of more sophisticated techniques.

Another study addressed the issue of labeled data scarcity in HAR by proposing a novel approach to combine multiple public datasets to create a generalized model that required less labeled data for effective fine-tuning on unseen domains Presotto et al. (2023). This model was trained specifically on data from waist-mounted devices and evaluated for its generalization ability across various datasets. The study revealed that focusing solely on waist-mounted data may not fully capture the variability of real-world scenarios, where devices may be placed at different body locations. Additionally, the performance variability across different datasets suggested that dataset-specific characteristics could significantly impact the effectiveness of the generalized model, and the advantage of pre-training might be limited when the target dataset diverges substantially from the other combined datasets.

An additional concern in the current literature is the prevalent use of multi-channel data for training models. While this approach benefits from rich, multi-dimensional inputs, it inherently limits the models' applicability to general kinematic signals that might be captured using simpler or different sensor configurations. This limitation raises questions about the scalability and flexibility of current HAR systems when deployed in real-world scenarios where sensor setups are not standardized and

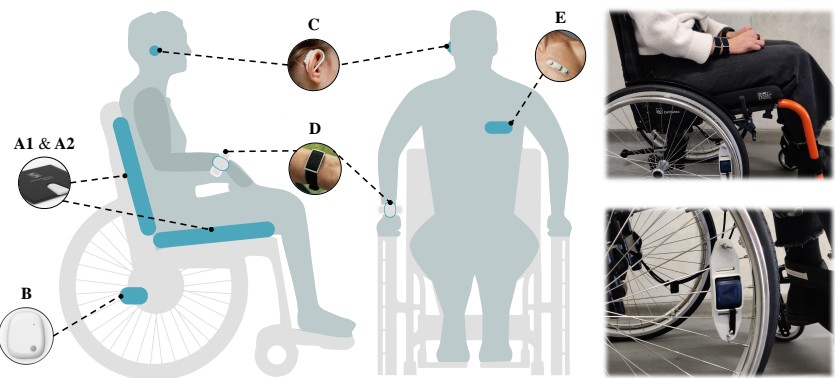

Figure 1: Composition of the sensory setup for the independent dataset. **A1 & A2:** Sensomative Wheelchair (pressure mats for backrest and bottom, respectively), **B:** mbientLab MMS+ (IMU), **C:** Cosinuss cmed°alpha (accelerometer), **D:** Vivalink Wearable ECG Monitor (accelerometer), **E:** Corsano CardioWatch 287-2 (accelerometer).

may vary greatly from one application to another (Gong et al., 2022; Suh et al., 2022; Sanabria et al., 2021; Al Hafiz Khan & Roy, 2022; Zhu et al., 2023; Alajaji et al., 2023; Qian et al., 2021; Varshney et al., 2022; Alinia et al., 2023; Lu et al., 2022; Faridee et al., 2022; Hu et al., 2023; Pavliuk et al., 2023; Haresamudram et al., 2022).

Despite the progress in this area, the HAR field yet faces notable limitations. Models frequently exhibit performance degradation when applied to domains that starkly differ from the training data, a phenomenon often observed in cross-domain applications Chen et al. (2019). Challenges such as overfitting and the disappearance of gradients in models trained on small datasets continue to hinder the reliability of these techniques Soleimani et al. (2022). Furthermore, the generalization of task-agnostic features often remains inadequate, leading to suboptimal performance when models are applied to new types of activities or different execution styles of the same activities Presotto et al. (2023). Another critical issue is the misclassification within activity clusters, where similar activities are often indistinguishable by the model, undermining its practical utility Kumar & Suresh (2023). Furthermore, the existing SOTA TL methods depend on a fixed sensor configuration or, at minimum, the same number of signals across all datasets, which limits their ability to leverage the diversity of available HAR datasets (Zhu et al., 2023; Alajaji et al., 2023).

## 3 MATERIALS AND METHOD

### 3.1 DATA

Eleven public HAR datasets were used to assess the generalization and classification efficacy of our proposed model: WISDM Kwapisz et al. (2011), MotionSense Malekzadeh et al. (2018), HHAR Stisen et al. (2015), REALWORLD Sztyler & Stuckenschmidt (2016), UniMiB SHAR Micucci et al. (2017), USC-HAD Zhang & Sawchuk (2012), MHEALTH Banos et al. (2014), PAMAP2 Reiss & Stricker (2012), WARD Yang et al. (2009), DSADS Barshan & Yüksek (2014), and RealDISP Baños et al. (2012). These datasets ensured a wide spectrum of human activities, signal modalities, and age ranges. The employed modalities included linear acceleration and angular velocity. Further, we validated the generalization of our model using our independent dataset (Our Previous Work, 2023). The sensory setup for the independent data shown in Figure 1 was designed to monitor activities of daily living in wheelchair users via modalities from accelerometers, gyroscopes, and pressure mat sensors.

Table 1, gives an overview of the datasets included in this study. The time series from all datasets were resampled to a sampling rate of 20Hz, adequate for monitoring daily human activities Ejtehadi et al. (2023). Our experiments utilized 20s sliding windows with a 50% overlap. A 20-s time frame is

Table 1: Overview of the open-source and the independent HAR datasets utilized in this study, detailing the number of activities, signals, subjects, and samples per activity for each dataset.

| Dataset | Num. of Activities | Num. of Signals | Num. of Subjects | Samples/Activity |
|---------|--------------------|-----------------|-------------------|------------------|
| DSADS (Barshan & Yüksek, 2014) | 19 | 30 | 8 | 48000 |
| HHAR (Stisen et al., 2015) | 6 | 6 | 9 | 557644 |
| Mhealth (Banos et al., 2014) | 12 | 15 | 10 | 11439 |
| Motionsense (Malekzadeh et al., 2018) | 6 | 6 | 24 | 94192 |
| PAMAP2 (Reiss & Stricker, 2012) | 12 | 18 | 9 | 32382 |
| REALDISP (Baños et al., 2012) | 33 | 54 | 17 | 8628 |
| REALWORLD (Sztyler & Stuckenschmidt, 2016) | 8 | 6 | 15 | 1140258 |
| UniMiB SHAR (Micucci et al., 2017) | 9 | 3 | 30 | 20923 |
| USCHAD (Zhang & Sawchuk, 2012) | 12 | 6 | 14 | 46861 |
| WARD (Yang et al., 2009) | 13 | 25 | 20 | 44169 |
| WISDM (Kwapisz et al., 2011) | 6 | 3 | 36 | 312579 |
| Independent Dataset | 8 | 39 | 20 | 26121 |

sufficient for most HAR applications. For convenience, we refer to the collective set of all eleven processed open-source datasets as datapool.

## 3.2 FRAMEWORK OVERVIEW

Figures 2, 3, and 4 provides a detailed depiction of the proposed framework for the study. To evaluate the generalizability and efficacy of our proposed method, we conducted a comparative analysis including 2 other baseline methods. Baseline method 1 (detailed in Figure 2), the standard practice, trained each dataset independently using a multi-channel FEN to process all signals concurrently,

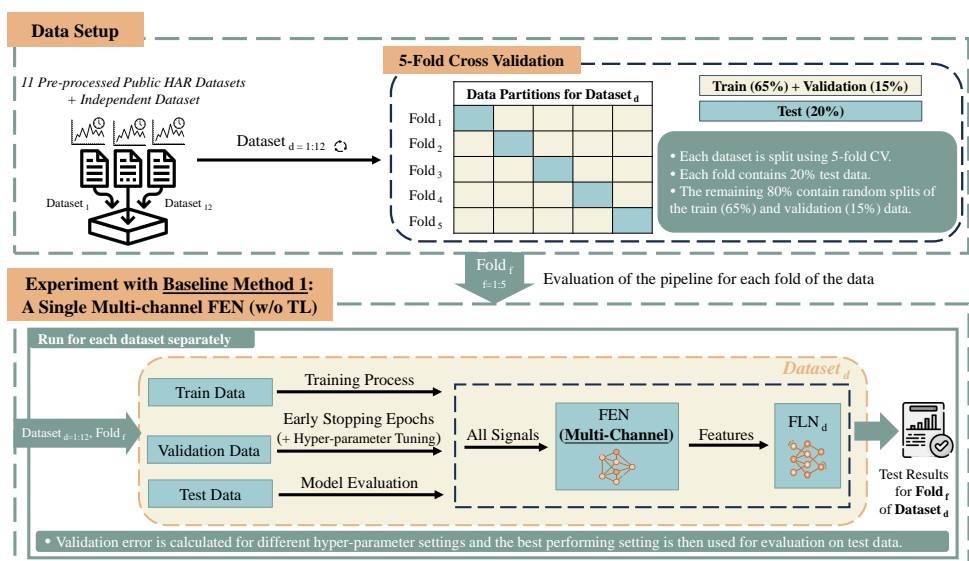

Figure 2: Flowchart of the experimental setup for baseline method 1, illustrating the feature learning process for human motion kinematic signals, with training performed independently for each dataset. This method utilizes a multi-channel FEN to learn features from all signals collectively. Indices $d$ and $f$ refer to the active dataset and fold respectively. Subscript $L$ indicates the number of available signals in the respective $dataset_d$

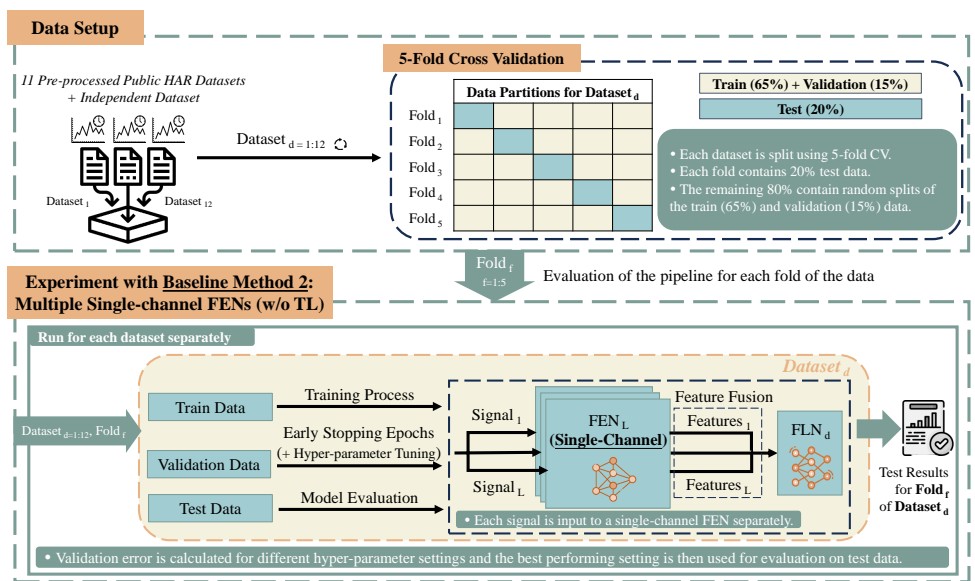

Figure 3: Flowchart of the experimental setup for baseline method 2, illustrating the feature learning process for human motion kinematic signals, with training performed independently for each dataset. This method utilizes a single-channel signal-specific FEN to learn features from the input signal. Indices $d$ and $f$ refer to the active dataset and fold respectively. Subscript $L$ indicates the number of available signals in the respective $dataset_d$

followed by the FLN for classification and feature learning. Baseline method 2 (detailed in 3), also a standard practice, utilized a distinct single-channel FEN for each signal, followed by the FLN for classification of the concatenated feature representation of the signals. Lastly, the GEAR-FEN method (detailed in Figure 4), structurally similar to method 2, employed one shared, single-channel FEN (GEAR-FEN) across all datasets and signals, aimed to achieve transferable feature representations from human motion signals. While the GEAR-FEN weights were continuously updated across all datasets and iterations, the FLN weights were only updated when training on the corresponding dataset within each iteration. The one-dimensional structure of GEAR-FEN would ensure its efficacy despite data heterogeneity, such as variations in the number of sensors and signals. This is despite the standard SOTA TL pipelines where a multi-channel FEN with fixed number of input signals is pre-trained and fine-tuned, which limits the pipeline to using the datasets with identical sensor and signal settings.

As discussed in section 3.1, all data underwent pre-processing for consistency. A 5-fold cross-validation (50-65% training, 15-30% validation, 20% test) ensured unbiased experimental results. To maintain the inter-subject variability and to avoid data leakage in the evaluation, we split the data by subjects. The performance was evaluated on the held-out test data of each fold without and with fine-tuning. In the former approach, the model from the final iteration was directly evaluated on each dataset. In the latter, the final model was further fine-tuned on the respective dataset before evaluation. The detail description of method 3 is provided in Algorithm 1.

### 3.3 NETWORK STRUCTURE & HYPER-PARAMETER TUNING

Inspired by Zhang et al. (2023a), the network architecture for all the baseline and GEAR-FEN methods incorporated CNN, ResBiLSTM, and an attention mechanism, effectively capturing the intrinsic patterns of human activity. For a detailed overview of the network structure, refer to Figure 5. The FEN architectures consisted of a 1-D CNN layer responsible for capturing low-level temporal dependencies, followed by ReLU activation and then max-pooling for dimensionality reduction. The FLN architectures consisted of a ResBiLSTM layer for learning long-term dependencies and temporal relationships from the dataset-specific features, followed by an attention mechanism and

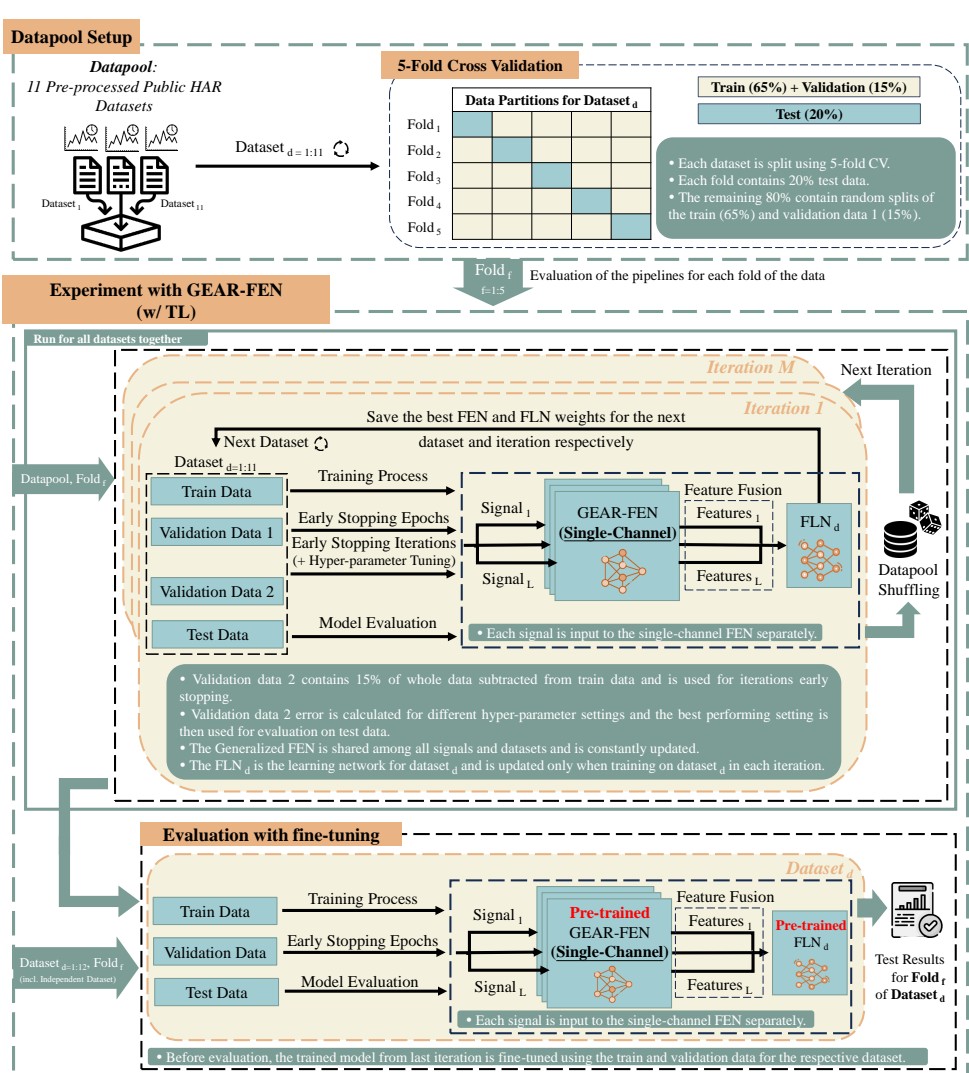

Figure 4: A Flowchart of the experimental setup employing TL through GEAR-FEN, illustrating the generalized feature learning process for human motion kinematic signals. This method utilizes a generalized single-channel FEN to learn features from any input signal. Indices $d$ and $f$ refer to the active dataset and fold respectively. Subscript $L$ indicates the number of available signals in the respective $dataset_d$

then a fully connected layer. The attention mechanism would allow the network to focus on the most relevant parts of the input sequence, improving its ability to identify significant patterns and filter out less important information. The fully connected layer would aggregate the attention-weighted features and produce the final classification output.

To ensure a fair comparison between the baseline methods and GEAR-FEN, hyper-parameter tuning was carried out for all three methods with the grid shown in Figure 5. Hyper-parameter tuning prioritized method 3 due to its access to a large data pool, supporting complex model structures without overfitting. The optimal hyper-parameters identified for GEAR-FEN method set the initial search space for baseline methods 1 and 2. Given GEAR-FEN method's extensive use of data, baseline methods 1 and 2 required a simplification of the model parameters to avoid overfitting. Moreover, since the FEN in the first baseline method employs a multi-channel approach, its output dimensions were the same as those of the single signals in the second baseline and GEAR-FEN

---

**Algorithm 1** GEAR-FEN Training Process

---

**Input:** HAR datasets $D_1, D_2, \ldots, D_N$, split into train, validation 1 (early stopping for epochs), validation 2 (early stopping for iterations), and test data
**Parameters:** Learning rate: $\eta$, Batch size: $n$
**Output:** Model weights $W$ for FEN and $W_d$ for $FLN_d$ for $D_d$

1: Initialize $W$
2: Initialize $W_d$ for $d = 1$ to $N$
3: **while** true **do**
4:     Shuffle the order of datasets $D_1$ to $D_N$
5:     **for** each dataset $D_d$ in shuffled order **do**
6:         **for** each signal $X$ in train data **do**
7:             Compute feature representation for each signal: $\hat{\Theta} = FEN(X; W)$
8:         **end for**
9:         Concatenate feature representations from all signals
10:        Update $W$ and $W_d$ based on early stopping criteria for epochs
11:     **end for**
12:     **if** Early stopping criteria for iterations **then**
13:         **break**
14:     **end if**
15: **end while**
16: **for** each dataset $D_d$ **do**
17:     Combine validation data 2 with train data for $D_d$
18:     Update $W$ and $W_d$ based on early stopping criteria for epochs
19: **end for**
20: **return** $W$ and $W_d$ for $d = 1$ to $N$

---

Table 2: Mean and standard deviation of f1-score performance comparison of the baseline methods, GEAR-FEN method, and SOTA performance (w/ or w/o TL). For the SOTA performances, the standard deviations are shown as n/a when not reported.

| Dataset | Average F1-Score | | | |
|---|---|---|---|---|
| | Method 1 | Method 2 | Method 3 | SOTA |
| DSADS | 0.80 ± 0.08 | 0.85 ± 0.04 | **0.89 ± 0.03** | 0.82 ± n/a (Su et al., 2022) |
| HHAR | 0.70 ± 0.04 | 0.70 ± 0.05 | **0.72 ± 0.04** | 0.52± n/a (Bock et al., 2022) |
| Mhealth | 0.83 ± 0.06 | 0.88 ± 0.05 | 0.92 ± 0.03 | **0.94 ± 0.04** (Suh et al., 2023) |
| Motionsense | 0.93 ± 0.05 | 0.95 ± 0.01 | **0.96 ± 0.01** | 0.92 ± n/a (Tahir et al., 2022) |
| PAMAP2 | 0.83 ± 0.10 | 0.93 ± 0.04 | **0.95 ± 0.03** | 0.86 ± n/a (Essa & Abdelmaksoud, 2023) |
| REALDISP | 0.83 ± 0.02 | 0.90 ± 0.01 | **0.92 ± 0.01** | 0.88 ± 0.17 (Suh et al., 2023) |
| REALWORLD | 0.73 ± 0.03 | 0.74 ± 0.02 | 0.76 ± 0.01 | **0.78 ± n/a** (Kwon et al., 2020) |
| UniMiB SHAR | 0.67 ± 0.04 | 0.72 ± 0.04 | **0.82 ± 0.06** | 0.77 ± n/a (Al-qaness et al., 2023) |
| USCHAD | 0.80 ± 0.06 | 0.81 ± 0.04 | **0.84 ± 0.05** | 0.83 ± n/a (Essa & Abdelmaksoud, 2023) |
| WARD | 0.89 ± 0.03 | 0.93 ± 0.02 | **0.95 ± 0.03** | 0.91 ± n/a (Yang et al., 2009) |
| WISDM | 0.80 ± 0.04 | 0.80 ± 0.05 | 0.82 ± 0.04 | **0.85 ± n/a** (Essa & Abdelmaksoud, 2023) |
| Independent Dataset | 0.56 ± 0.07 | 0.73 ± 0.07 | **0.85 ± 0.04** | – |

methods after the concatenation of features. Further, for a fair comparison between the feature extraction pipelines, the FLNs for all methods shared the same hyper-parameters. The batch size was set at 64 to effectively manage computational resources and model performance.

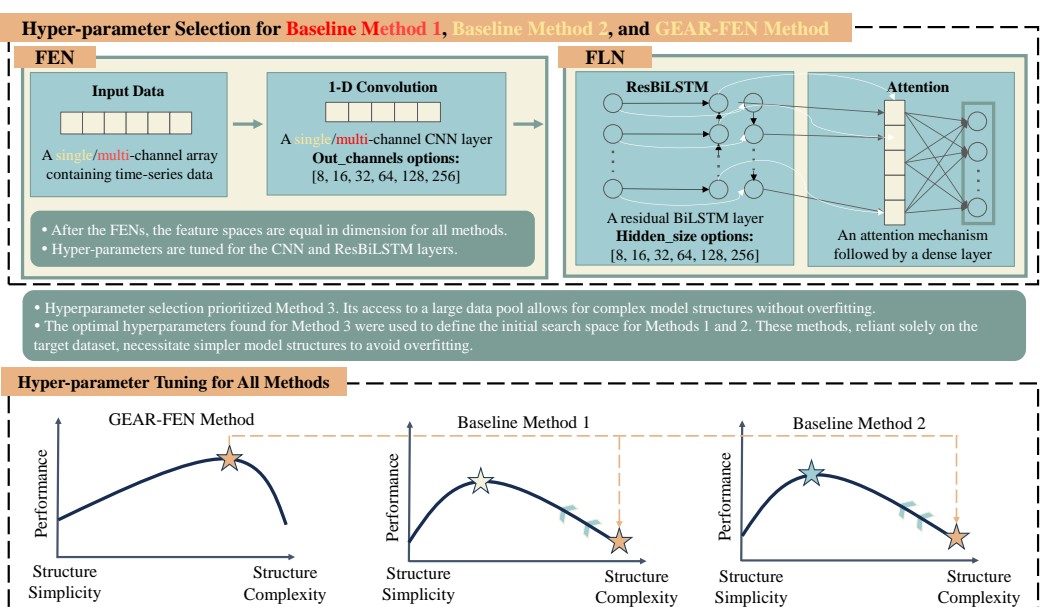

Figure 5: Network architectures and hyper-parameter grids for the methods used in this study. GEAR-FEN method's access to ample data allowed for increased complexity without overfitting. The optimal settings for GEAR-FEN method defined the initial hyper-parameter grid space for baseline methods 1 and 2.

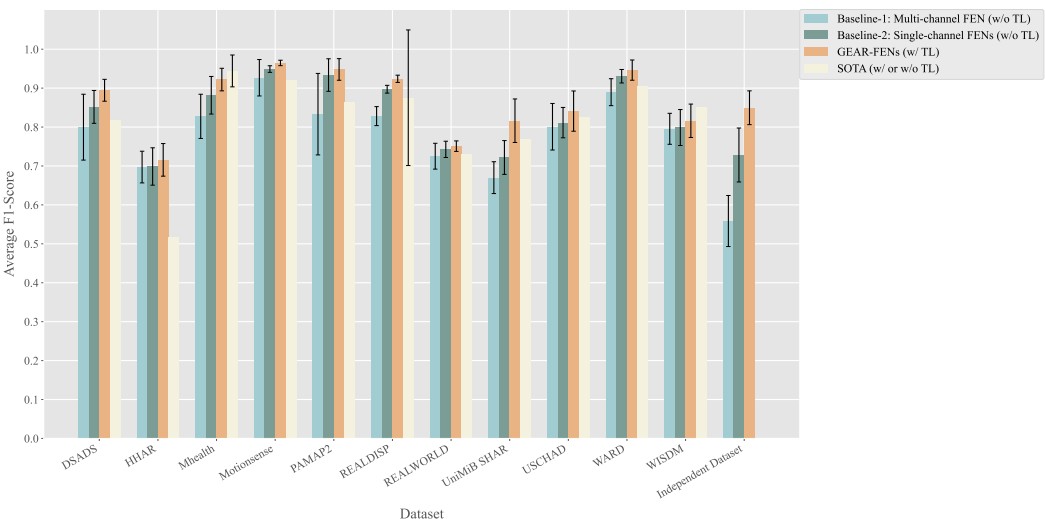

Figure 6: Barplot visualization of the mean and standard deviation of f1-score performance comparison of the baseline methods, GEAR-FEN method, and SOTA performance (w/ or w/o TL). For the SOTA performances, the standard deviations are visualized when reported.

## 4 EXPERIMENTS AND RESULTS

Table 2 presents the mean and standard deviation of f1-scores, averaged across all five experimental folds, for all methods and datasets in this study. The table also compares the baseline and GEAR-FEN performances with the SOTA performance for each dataset. Further, Figure 6 the barplot depicts

the barplot representation for the comparative f1-scores of the baseline and GEAR-FEN methods, alongside the SOTA performances.

The results suggest that the GEAR-FEN method consistently achieves better or comparable results in terms of f1-score, always outperforming the other baseline methods and often outperforming the SOTA. In 9 out of 12 datasets, GEAR-FEN achieves the highest average f1-score. For instance on DSADS, GEAR-FEN achieves a score of 0.89 ± 0.03, significantly better than baseline method 1 (0.80 ± 0.08), baseline method 2 (0.85 ± 0.04), and SOTA (0.82). On UniMiB SHAR, GEAR-FEN outperforms the baselines with a score of 0.82 ± 0.06, compared to baseline 1 (0.67 ± 0.04) and baseline 2 (0.72 ± 0.04), and surpasses SOTA (0.77). Similarly, GEAR-FEN consistently outperforms on datasets like PAMAP2, REALDISP, and WARD, with higher mean f1-scores and relatively low standard deviations, indicating robust performance. In some cases, like Mhealth and WISDM, the GEAR-FEN method's performance is slightly below SOTA, but it still shows competitive results with a narrower margin for Mhealth. In the case of the independent dataset, the GEAR-FEN method (0.85 ± 0.04) outperforms the baseline methods (0.56 ± 0.07 and 0.73 ± 0.07) by a considerable margin. Also the standard deviation has a narrower margin which shows the robustness of GEAR-FEN against the baseline methods.

## 5 Discussion

The results demonstrate that the GEAR-FEN method consistently outperforms the baseline methods and often exceeds SOTA performance across majority of the investigated HAR datasets. GEAR-FEN demonstrates strong generalization, particularly in more challenging datasets with limited sample size, and shows greater robustness, as indicated by lower variance in f1-scores. Even in cases where it does not surpass SOTA, GEAR-FEN remains competitive, highlighting its effectiveness in feature learning for human motion signals. The observed improvements in test accuracy across iterations for a variety of datasets, substantiate the effectiveness of our model and training approach. Importantly, the ability of our model to generalize across such a wide range of HAR datasets supports the hypothesis that kinematic signals e.g., linear acceleration and angular velocity and motion-related kinetic modalities such as those in pressure mats exhibit similar patterns, which GEAR-FEN can capture.

Moreover, the superior performance of the proposed models on an independent dataset indicates that the feature representation learned by GEAR-FEN is highly transferable for human motion patterns. This transferability has resulted in significant improvements when applied to datasets with small sample sizes, and with new populations, sensor modalities, and sensor locations, as demonstrated by the independent dataset's performance.

Additionally, the results show that feature learning using multiple single-channel FENs consistently outperforms feature representation from a multi-channel FEN, highlighting the structural superiority of GEAR-FEN compared to SOTA transfer learning networks. It's also important to note that GEAR-FEN's generalized single-channel structure addresses issues related to incompatible signal counts and sensor locations in transfer learning. This design enables highly customizable fine-tuning of the model, making GEAR-FEN adaptable to diverse datasets and sensor configurations

## 6 Conclusion

In this paper, we demonstrated a pipeline for automated feature representation of the kinematic signals related to human motion. This pipeline can be used for transferring the signals into a domain of representative features, capable of distinguishing between human activities. The generalized feature extraction pipeline, combined with feature fusion models, contributes to HAR research, outperforming most of the benchmark models in the field.

Further, the proposed activity classification pipeline outperformed most of the SOTA scores for different HAR datasets. It is recognized that SOTA models in HAR heavily rely on large and diverse datasets, posing significant challenges in data acquisition. Our research addresses these issues by focusing on generalizing across all kinematic signals. While aiming for a model that generalizes across all kinematic signals, we acknowledge that some datasets show lower accuracy compared to certain SOTA models. This trade-off is acceptable in our goal to create a universally adaptable system. It is noteworthy that the proposed GEAR-FEN pipeline can be further fine-tuned using

the HAR datasets with more inherent similarities to the target dataset. The customized fine-tuning can aid the model in capturing the local and intricate patterns and features specific to a set of activities or populations. Moreover, validation of the independent dataset confirms the effectiveness of GEAR-FEN's generalization capability.

Our research revealed that the feature representation pipeline exhibits superior performance compared to the benchmark deep learning models when applied to an independent dataset. This dataset encompassed a diverse population (wheelchair users) and incorporated various sensor modalities, including pressure distribution. The findings suggest that the feature learning pipeline effectively captures the essential characteristics of motion from the signals, potentially enhancing classification performance when integrated with a classification model. The balance between broad applicability and dataset-specific precision highlights the potential of our models in diverse real-world applications and sets a direction for future research in HAR technology.

In our study, we acknowledge a limitation of the feature learning pipeline's applicability to kinematic signals associated with human motion. Further, we acknowledge that the GEAR-FEN methodology has been validated only with a specific neural network structure (comprising CNN, ResBiLSTM, and attention mechanism). The pipeline should be further validated for different neural network structures. In forthcoming research, we intend to also employ the Local Interpretable Model-Agnostic Explanations (LIME) technique to elucidate the features extracted by the pipeline. This will help assess whether the feature representation pipeline accurately captures features relevant to the target output. Given the pipeline's capability to transform any signal into the feature domain, conducting an activity-wise analysis of these features will enable us to delineate the contribution of each input signal modality to the recognition of various activities. Furthermore, it should be investigated whether the generalized feature representation pipeline can be extended to physiological signals with repeatable patterns, such as heart rate and blood pressure.

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
