# OpenReview forum: "GEAR-FEN: Generalized Feature Representation for Kinematic Human Activity Recognition"
_ICLR.cc/2025/Conference — ICLR 2025 Conference Withdrawn Submission_

### Official Review · Reviewer_Bmxw · 2024-10-17

**Soundness:** 1
**Presentation:** 2
**Contribution:** 1
**Rating:** 1
**Confidence:** 5

**Summary:**

This paper proposes a transfer learning method for human activity recognition that transforms kinematic motion signals into a generalized feature space. The author used CNN, ResBiLSTM, and an attention mechanism to create a deep learning model and train it with 11 different HAR datasets to improve the classification performance.
The authors also compare the proposed method with SOTA and achieve better performance in terms of F1-score.

**Strengths:**

This paper introduces a deep learning method using a deep learning model based on convolutional neural networks (CNN),residual-bidirectional long short-term memory (ResBiLSTM), and an attention mechanism.
The author tested their proposed method using 11 different HAR datasets.
Through performance comparisons using an independent dataset that included a different demographic and a range of kinematic modalities for eight activities, with 26121 samples per activity, the authors demonstrated the generalizability of the proposed approach.

**Weaknesses:**

Its weaknesses are listed below:
1. The proposed method integrates multiple existing methods.  Some existing similar methods are in the community.  The innovation of the proposed method is not significant in the community.
2. The research gaps should be explained in the manuscript.
3. The authors should summarize the drawbacks of the existing relevant literature and review some recent state-of-the-art HAR literature.
4. Please provide some necessary mathematical reasonings in the manuscript.
5. The authors should explain the differences between the proposed method and existing algorithms.
6. The experimental analysis is somewhat weak:
a) The authors fail to compare the proposed method with recent algorithms, especially after 2023.
b) Computational complexity should be analyzed and compared in the manuscript.
c) An ablation study is missing in the manuscript.
7. The authors provide a case study to help readers understand the proposed method.

**Questions:**

1. What are the innovations of the proposed method?
2. What are the improvements of the proposed method compared with similar studies?
3. What is the significance use to combining CNN, ResBiLSTM, and an attention mechanism?
4. How attention mechanism is working in this proposed architecture?
5. How many layers are involved in the attention mechanism?
6.  What are the limitations of the proposed method? How will you solve them?
7. Is the independent, unseen dataset real-time?
8. What will be the performance of the real-time dataset?

---

### Official Review · Reviewer_YDhi · 2024-11-02

**Soundness:** 2
**Presentation:** 2
**Contribution:** 2
**Rating:** 3
**Confidence:** 4

**Summary:**

Human activity recognition (HAR) is largely limited by the lack of high quality labelled data. Thus, transfer learning on pre-trained models is a commonly employed approach. However, due to the heterogeneity of HAR datasets, the performance the existing methods fail when the data distributions are different between the source and target domain.

This manuscript proposes a generalised feature representation framework to HAR that aims at providing a uniformed pre-training framework. The authors aggregated 11 public HAR datasets for supervised pre-training to obtain a generic feature extractor. Another key feature of this work is a dataset-specific feature learning network that will tailor the feature space for each dataset during fine-tuning.

The authors reported competitive performance on public HAR benchmarks against two other variations of the pre-training pipeline. They also contextualised their performance against the benchmark performance from the literature. Finally, they tested their pre-trained model on an independent test dataset to showcase its external validity.

**Strengths:**

* This paper tackles an important research question in HAR. Often there is a mismatch between the source and target data distributions, and thus this work provides another tool to the community to address the data heterogeneity.
* The list of public datasets included is comprehensive.
* The proposed solution is an interesting attack on developing dataset-specific feature extractor on top of a generic feature encoder.

**Weaknesses:**

* The included baseline models are limited as only variations of proposed model was used. Would be good to include other baseline models, at least models that are fully supervised.
* It is not clear how the evaluation was done. If my understanding is correct, the pre-training on the pool of public datasets was done using supervised pre-training. After obtaining a dataset-specific feature learning network on one dataset, how does the fine-tuning work on the same dataset with the same set of labels?
* Even though developing some dataset-specific feature extractor on top of a generic feature extractor is a neat idea, I don’t see how using supervised pre-training on the same dataset with the same labels, then do the fine-tuning is any different from simply training a network from scratch with the same labels. If my understanding is wrong on what the authors actually did, I would be happy to revisit my rating during the rebuttal.
* Only average F1 scores were reported when comparing performance across baseline models, which is insufficient for provide the superiority of the proposed work. More ablation studies including how the model perform under different distribution shifts, data distribution regimes are needed.

**Questions:**

1. In text citations not appropriately used e.g. L56, L76
2. L243 add (detailed in 3) -> (detailed in Figure 3)
3. Can the authors remove the unnecessary details for their data partitioning, optimisation and hyper-parameter strategies. The same content can be stated once but not every time in every baseline model description and figure?
4. More details on the model architecture?
5. How is the fine-tuning done?
6. Fold f is listed in every figure, is there anything special that's done in each fold? If not, please remove to aid readability.

---

### Official Review · Reviewer_ojhd · 2024-11-03

**Soundness:** 2
**Presentation:** 2
**Contribution:** 2
**Rating:** 3
**Confidence:** 5

**Summary:**

This paper introduces GEAR-FEN, a transfer learning-based approach for generalizing feature representation for human activity recognition using wearable sensor data. The authors claim that GEAR-FEN can handle data scarcity and achieve robust feature representation by using multiple public HAR datasets. The approach combines a CNN, ResBiLSTM, and attention mechanisms to improve feature extraction representations. The proposed method was evaluated across 11 HAR benchmark datasets compared to SOTA methods.

**Strengths:**

+ The challenge of generalizing HAR models across multiple datasets with limited training data is a significant issue in this field. the authors tried addressing the limitations of data-specific models with a feature extraction network for generalization.
+ Evaluating across 11 HAR datasets with several SOTA methods provides a comprehensive benchmarking approach. This shows the proposed method generalizes across datasets with different data distributions and activity types.

**Weaknesses:**

- The main concern with this paper is its lack of novelty. while the use of transfer learning for generalization across HAR datasets is interesting, it does not introduce a fundamentally new approach or architecture. The proposed combination of CNN, ResBiLStM, and attention mechanisms is a commonly adopted approach in recent HAR literature.
- The authors used a 20-second sliding window across datasets, which raises concerns about the rigor and comparability of their experiments. Most HAR literature typically uses much shorter windows (e.g., 2-5 seconds) for real-time applications and responsiveness, and some of the datasets in this paper likely follow this convention. The selection of the 20-second window could potentially inflate performance due to capturing extended contextual information, which may not align with previous studies.
- While GEAR-FEN is tested across multiple datasets, the methodology does not introduce new insights into feature extraction for HAR. It seems to an incremental work in this area.

**Questions:**

- Can the authors provide further justification for using a 20-second window length across all datasets? Do authors observe performance variations when experimenting with shorter window lengths like 2 seconds to 5 seconds?
- Please highlight the novelty and any unique aspects of GEAR-FEN's architecture or approach that differentiate it from existing HAR methods.

---

### Official Review · Reviewer_SoUu · 2024-11-04

**Soundness:** 2
**Presentation:** 2
**Contribution:** 2
**Rating:** 3
**Confidence:** 3

**Summary:**

The study proposes a generic framework for learning features across several datasets and tasks for human activity recognition. The proposed Generalized Activity Recognition Feature Extraction Network outperformed several baseline and previous studies.

**Strengths:**

A strength of the paper is that it tests the model across 11 datasets and evaluates it on a new dataset.
The paper also presents extensive evaluations and comparisons with several baselines, which I have some reservations about.

**Weaknesses:**

1) The dataset division is not comparable with previous studies. Previous work could test the models on unseen subjects or use agreed-upon divisions. I understand that applying the same strategy for evaluations across the 11 datasets is convenient and understandable, but when compared to other work, it is recommended to follow the standards.

2) The authors claimed that the datasets are extensive (in terms of modalities) but only include acceleration and angular velocity.


3)The proposed architecture is rather vague and brute force. The paper discusses aspects related to its complexity and performance in very generic terms without concrete numbers and results. E.g., Figure 5 presents three figures to compare the two baseline methods and the proposed approach and shows vaguely that the proposed model's complexity can increase without compromising the performance, unlike the proposed model.

4) The way to handle the signals with a single network does not seem novel. Other reviewers could better validate this idea.

5) The paper motivates the idea of transfer learning, without any support. The presented results are in bulk, and there are not comparisons and in depth analyses of the effectiveness of the transfer learning proposed by the paper.

**Questions:**

1) When citing a previous work, authors can just cite it with actual reference and refrain from improper referencing.

2) The datasets' setup does not need to be repeated for each figure, nor even depicted. It is quite clear how the setup is conducted, and a textual explanation is sufficient, which can save space if needed.

3) The baselines are not sufficiently described or visualized. The space used to depict the train/validation/text setup (repeated three times). What is the baseline method 2? How is it different from the proposed method? These architectures are not clear.

---

### Note · Authors · 2024-11-29

I have read and agree with the venue's withdrawal policy on behalf of myself and my co-authors.